# HANDWRITTEN TEXT RECOGNITION ADAPTATION FOR LOW-RESOURCE LANGUAGES: A CASE STUDY ON HISTORICAL LATIN MANUSCRIPTS

## ABSTRACT

Handwritten Text Recognition (HTR) remains a challenging task in document digitization, particularly for historical manuscripts written in low-resource languages such as Latin. In this paper, we focus on recognizing Latin texts from 16th–18th century manuscripts, which exhibit a wide range of handwriting styles. To address this, we propose AdapterTrOCR, a modular extension of the TrOCR model that incorporates two adapter modules: one for historical language adaptation and another for handwriting style adaptation. This architecture enables a robust transition from a modern English HTR model to one specialized in historical Latin. Given the limited availability of annotated data, we also explore Handwritten Text Generation (HTG) as a data augmentation strategy. Our results show the effectiveness of modular adaptation and synthetic data in improving HTR performance, achieving reductions in character error rate (CER) by 13.33% to 35.65% and word error rate (WER) by 8.56% to 27.72%.

## 1 INTRODUCTION

Text recognition is an ongoing research challenge in the context of document digitization, aiming to extract textual content from real-world and visually complex scanned documents. It encompasses a variety of tasks, ranging from printed text recognition to scene and handwritten text recognition. While printed text recognition typically involves clean layouts and consistent fonts in scanned documents, the other subtasks present greater challenges. Scene text recognition must handle complex backgrounds, varying lighting conditions, distortions, and font diversity (Xu et al., 2024; Du et al., 2025; Zhao et al., 2024), whereas the variability in individual handwriting styles, slant, and spacing complicates handwritten text recognition (HTR) (Li et al., 2023; 2025; Gu et al., 2025). Additionally, alternative approaches for text recognition address specialized content such as mathematical expressions and tables (Zheng et al., 2021; Kishor et al., 2023; Wan et al., 2024; Loitongbam & Middleton, 2025).

In this paper, we focus on HTR as a means of digitizing historical manuscripts. These manuscripts, belonging to a well-known European library, are written in Latin and date from the 16th to 18th centuries. They consist of student notes that exhibit a wide variety of handwriting styles and cover diverse domains, from logic to physics[1].

HTR can be implemented as a two-step process: first, detecting individual text lines (hereafter referred to as line images), and then recognizing the text within each line (Figure 2 in the Appendix shows an example of how scanned manuscript pages are segmented into line images). Recent end-to-end methods unify line detection and text recognition (Wigington et al., 2018; Kwon et al., 2023; Mao et al., 2024; Hamdi et al., 2025). As, our preliminary experiments showed that, in our case, the end-to-end approach tend to be more error-prone and more difficult to interpret and debug than the two-step pipeline, we therefore adopted the latter. While line detection can be addressed effectively by fine-tuning an object detection model, line-level HTR remains the more challenging task due to limited training data for Latin scripts and the highly variable handwriting in historical manuscripts. In this work, we focus exclusively on the second step, hereafter called line-level HTR.

---

[1]To preserve anonymity, we will provide more details about the collection of manuscripts upon acceptance.

To recognize the text within line images, we fine-tune TrOCR (Li et al., 2023), a well-established encoder-decoder model pretrained for HTR in English, on a Latin dataset. This transfer learning approach is commonly used for languages with fewer HTR resources than English (Ströbel et al., 2022; Lauar & Laurent, 2024), as it facilitates the learning process by transferring the HTR knowledge encoded in the English-trained TrOCR model.

Since the simple fine-tuning is not sufficient, we imagine the transition from a modern English TrOCR to a historical Latin TrOCR[2] as a linear equation. This equation is realized by integrating two adapter modules into the TrOCR architecture, resulting in the proposed AdapterTrOCR model. The first module performs historical language adaptation, transforming the representations learned from English into a space suitable for historical Latin handwritten text. The second module focuses on style adaptation, allowing the model to adjust to the specific handwriting styles found in the manuscripts. Both modules are trained on dedicated datasets and then integrated into the AdapterTrOCR model, which is subsequently fine-tuned on the Latin corpus. Although AdapterTrOCR is designed for Latin, its modular architecture makes it easily adaptable to other languages.

Due to limited data availability, we also explore handwritten text generation (HTG) as a form of data augmentation for specific handwriting styles. While HTG typically performs well on handwriting styles that are well represented in the training data and where data augmentation is less critical, we propose a solution that also benefits underrepresented handwritten styles.

The contributions of our work are summarized as follows:

1. We develop AdapterTrOCR, a new model for historical Latin HTR by decomposing the components needed to transition from a modern English HTR model to one tailored for historical Latin manuscripts handwritten in specific handwriting styles.

2. We propose DiffLine, a new HTG-based data augmentation method, and demonstrate its effectiveness for HTR, particularly in the case of underrepresented handwriting styles.

The remainder of the paper is organized as follows. Section 2 reviews related work on HTG and HTR. Section 3 introduces our methodology, including AdapterTrOCR and DiffLine, which are evaluated in Section 4. Finally, Section 5 presents our conclusions, limitations, and directions for future research.

## 2 RELATED WORK

**Handwritten Text Generation.** As in many image generation tasks, adversarial training has made a significant contribution to the generation of photorealistic images containing handwritten text. Generative Adversarial Network (GAN)-based methods range from using only textual input for conditioning (Alonso et al., 2019; Fogel et al., 2020; Zdenek & Nakayama, 2021) to incorporating both text and handwriting style as conditioning signals (Kang et al., 2020; 2021; Mattick et al., 2021; Bhunia et al., 2021; Pippi et al., 2023; Gan et al., 2022; Wang et al., 2025; Hoai Nam et al., 2025). More recently, diffusion models have become dominant in this domain due to their ability to produce higher-quality image samples than GANs (Dhariwal & Nichol, 2021).

While diffusion-based models for HTG typically rely on a U-Net architecture (Ronneberger et al., 2015) for denoising, they mainly differ in how the text and style conditions are encoded and integrated into the diffusion process. Zhu et al. (2023), Mayr et al. (2024) and Dai et al. (2024) use a transformer decoder to merge the style and text embeddings into a single representation, which is then provided to the U-Net as a unified conditioning signal. Gui et al. (2023) propose an HTG model that follows the InstructPix2Pix framework (Brooks et al., 2023), where the text condition is represented as a glyph image displayed in a standard font and concatenated with the input image, while the style embedding guides the U-Net's noise prediction. This approach is further extended by Ding et al. (2023) by introducing a filtering module that discards synthetic text images with low HTR scores.

---

[2]In the context of this paper, Historical Latin TrOCR refers to a TrOCR model trained to recognize Latin texts in historical manuscripts dating from the 16th to 18th centuries.

Considering that style is a global property affecting the entire image and text is a sequential and spatial signal, WordStylist (Nikolaidou et al., 2023) injects the style embedding into the U-Net by summing it with the timestep embedding, while the text condition is incorporated via cross-attention. The method is further refined in DiffusionPen (Nikolaidou et al., 2024), which employs a CANINE-C text encoder (Clark et al., 2022) and a MobileNetV2 (Sandler et al., 2018) for style encoding.

A recent alternative to adversarial and diffusion-based models is presented by Pippi et al. (2025), where the authors propose an autoregressive transformer-based approach. The method reconstructs input text images without background, aiming to enhance the clarity and quality of text rendering in the generated outputs.

**Handwritten Text Recognition.** Text recognizers for line images typically rely on convolutional neural networks (CNNs) to learn spatial patterns (Puigcerver, 2017; Shi et al., 2017; Puigcerver, 2017; Wigington et al., 2018; Ahlawat et al., 2020; Yousef & Bishop, 2020; Chaudhary & Bali, 2022; Coquenet et al., 2021), or incorporate attention mechanisms such as transformer-based blocks (Wang et al., 2020; Kang et al., 2022; Li et al., 2022).

More recently, Li et al. (2023) proposed TrOCR, an encoder-decoder architecture in which the encoder is based on the BEiT model (Bao et al., 2021), and the decoder is initialized with the weights of a RoBERTa model (Liu et al., 2019). Rather than using the full Transformer-based encoder-decoder structure, Li et al. (2025) employ only the Transformer encoder, initialized with the weights of a Vision Transformer (ViT) (Dosovitskiy et al., 2021), for text recognition within the line images. This approach includes a convolutional-based feature extractor and utilizes the Sharpness-Aware Minimization (SAM) optimizer (Foret et al., 2021). Alternatively, Fujitake (2024) propose an HTR model based on a Transformer decoder initialized with a GPT model (Radford et al., 2019), while image patches are represented using the patch embedding technique described by Dosovitskiy et al. (2021).

Unlike the aforementioned models, which adopt writer-independent approaches, Wang & Du (2022) embed handwriting style into a vector representation and integrate it into a CNN model to enhance text recognition performance. Another writer-specific personalization method is presented by Gu et al. (2025), where learnable writer-specific vectors are combined with input line images through spatial concatenation or padding. While this approach is similar to our proposed HTR model, MetaWriter (Gu et al., 2025) applies personalization only at the level of the convolutional layer due to the limitations of the padding-based implementation. In contrast, our style adapters can be seamlessly integrated throughout the entire network, allowing for a deeper and more holistic influence of the handwriting style on the HTR model.

## 3 PROPOSED METHODOLOGY

We begin by introducing the proposed AdapterTrOCR model, followed by a description of the HTG-based data augmentation strategy used to expand the training data for AdapterTrOCR.

### 3.1 ADAPTERTROCR FOR HANDWRITTEN TEXT RECOGNITION

The HTR model we propose is based on the TrOCR architecture (Li et al., 2023), which employs an encoder-decoder design. Our choice of TrOCR over other visual-language models stems from the fact that it is already pretrained for the HTR task. More details about this in Appendix A.1 In this paper, our task is to adapt TrOCR, pretrained for English HTR on modern datasets, to handle historical Latin manuscripts written in diverse handwriting styles. To this end, we introduce AdapterTrOCR, which incorporates two modules that independently adapt TrOCR for historical Latin HTR and for the recognition of text in a specific handwriting style.

The adaptation is carried out in two steps (Fig. 1a). First, we train the two adapters on specific datasets and tasks, which will be described in detail below. These adapters utilize only the decoder component of TrOCR, as some of the training tasks involve only the language modality, which is handled exclusively by the decoder. In the second step, we integrate the trained adapter weights into the full TrOCR architecture and fine-tune the entire model on a Latin-based dataset. While the goal of our project is to transcribe our Latin collection of old manuscripts, the proposed AdapterTrOCR can be easily applied to other languages as well. More details in Appendix A.2.

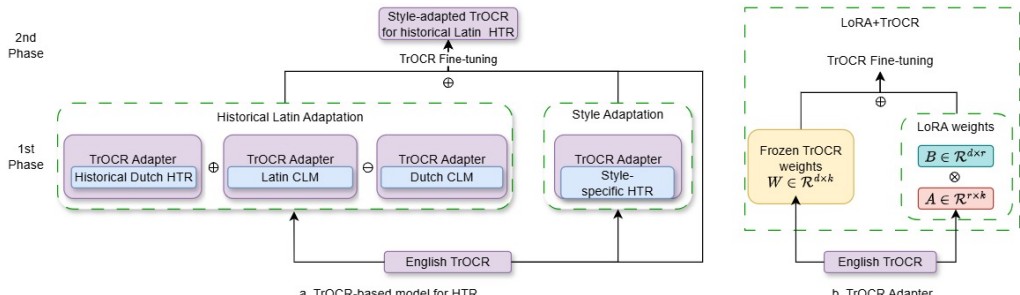

Figure 1: The historical Latin and style adaptations of the TrOCR model for HTR (a) and the TrOCR adapter implemented with LoRA (b).

Given a TrOCR layer $h_1 = W h_0$, where $h_0 \in \mathbb{R}^k$ is the input, $W \in \mathbb{R}^{d \times k}$ is the initial TrOCR weight, and $h_1 \in \mathbb{R}^d$ is the output of the layer, the adapters are trained to learn new weights $W' \in \mathbb{R}^{d \times k}$ that are integrated into the layer as follows: $h_1 = (W \oplus W')h_0$, where $\oplus$ represents the element-wise addition. To reduce the training overhead of the adapters, we obtain the matrix $W'$ using a low-rank decomposition implemented via LoRA (Hu et al., 2022). The layer is adapted as follows:

$$h_1 = (W \oplus W')h_0 = (W \oplus BA)h_0 \tag{1}$$

where $B \in \mathbb{R}^{d \times r}$ and $A \in \mathbb{R}^{r \times k}$ are the low-rank decomposition matrices of $W'$ and $r \ll min(d,k)$. During the training of the adapters, all TrOCR parameters are frozen except for the $B$ and $A$ weights (Fig. 1b). After the adapters' training, the LoRA weights of the historical Latin adapter $W'_{hl}$ and the LoRA weights of the style adapter $W'_s$ are integrated into a TrOCR layer as follows:

$$h_1 = (W \oplus \lambda_{hl} W'_{hl} \oplus \lambda_s W'_s)h_0 \tag{2}$$

This formulation relies on the compositionality rules proposed by Zhang et al. (2023), which enable a model to transition from a task to another one by performing arithmetic operations on the parameters of adapters trained for individual tasks. All TrOCR parameters, including the weights of the adapters, are subsequently fine-tuned on a Latin-specific dataset[3].

**Historical Latin adaptation**  To define the historical Latin adaptation, we require the TrOCR model to distinguish between "task ability" and "language ability". "Task ability" refers to the model's capacity to perform historical HTR using a proxy language, while "language ability" denotes the adaptation from the proxy language to Latin. This approach involves first preparing the model to handle historical handwriting in the proxy language ("task ability"), and then removing the difference between this language and the target language, which in our case is Latin.

To define the "task ability", we build an adapter that redirects TrOCR from modern HTR—obtained through pretraining on modern English handwriting datasets, as discussed by Li et al. (2023)—to the task of historical HTR. This adapter is a TrOCR-based decoder trained as a parameter-efficient module (PEM) using LoRA on the *VOC and notarial deeds dataset* (Keijser, 2024). The choice of this dataset is motivated by its inclusion of manuscripts from a similar time period as those in our collection of manuscripts (from the 16th to 18th century). The proxy language is Dutch, as it is the language used in the VOC and notarial deeds manuscripts. While languages more closely related to Latin do exist, we retain Dutch as the proxy language because, to the best of our knowledge, the *VOC and Notarial Deeds dataset* is the largest and highest-quality human-annotated HTR dataset available for the historical period relevant to our collection. Further details are provided in Appendix A.4.

To extract the "language ability", we rely on the assumption that the difference between two languages can be captured by the difference between the weights of a model trained on a proxy task

---

[3]Even though TrOCR is pre-trained for HTR, we do not freeze its encoder because the visual characteristics of the pre-training modern data differ from those of our historical manuscripts. More details are provided in Appendix A.3.

in one language and the weights of a model trained on the same task in the second language (Zhao et al., 2025; Ansell et al., 2022; Zhang et al., 2025). Based on this assumption, we define two adapters—both TrOCR-based decoders—and train them separately as PEMs using LoRA on Dutch and Latin. Since there is no restriction on the choice of the proxy task (Zhao et al., 2025; Ansell et al., 2022; Zhang et al., 2025), we employ a self-supervised task, such as causal language modeling (CLM), to train the adapters.

Once the adapters are trained, we rely again on the compositionality rules discussed by Zhang et al. (2023) and define the LoRA weights $\boldsymbol{W}'_{hl}$ as:

$$\boldsymbol{W}'_{hl} = \boldsymbol{W}'_h \oplus \lambda_l(\boldsymbol{W}'_l \ominus \boldsymbol{W}'_d) \tag{3}$$

where $\boldsymbol{W}'_h$ denotes the LoRA weight learned by the task adapter for historical HTR, $\boldsymbol{W}'_d$ and $\boldsymbol{W}'_l$ represent the LoRA weights associated with the Dutch and Latin language adapters, respectively, and $\ominus$ is the element-wise subtraction.

**Style adaptation**  Similar to historical Latin adaptation, style adaptation is achieved by training a TrOCR-based decoder as a PEM on a subset of images containing text written in a specific handwriting style (extracted from the same manuscript). In this work, we assume that all line images extracted from a given manuscript are written by a single author and therefore exhibit a consistent handwriting style. To obtain sufficient data for training the style adapter, we augment the real annotated line images from a manuscript with synthetic line images generated in the corresponding handwriting style. This step is particularly important for underrepresented manuscripts that do not contain sufficient annotated line images in the training data. After training the style adapter, the LoRA weight $\boldsymbol{W}'_s$ is integrated into the TrOCR architecture, as indicated in Eq. 2.

## 3.2 Handwritten Text Generation for Data Augmentation

Since the annotated data is limited, we propose an HTG module to augment our dataset. The goal of integrating HTG into our framework is to enrich small subsets of line images handwritten in a specific style by generating synthetic lines that preserve the same handwriting characteristics. These synthetic line images are used to augment the training data for both AdapterTrOCR and the style adapter.

To achieve HTG-based data augmentation, we define a diffusion-based model capable of generating synthetic data conditioned on both style and text. The proposed model, which we call DiffLine, is based on DiffusionPen (Nikolaidou et al., 2024), the current state-of-the-art in HTG. As previously noted, DiffusionPen, built on top of WordStylist (Nikolaidou et al., 2023), has the advantage of treating handwriting style as a global condition that influences the entire line image, while modeling text as a spatial condition. Building upon DiffusionPen, we integrate dual classifier-free guidance for both conditions and enhance the style encoder by employing a more robust training regime than the one originally used in DiffusionPen.

**Diffusion Models with Dual Classifier-free Guidance for Style and Text Conditions**  Diffusion models are a class of generative models that produce data by reversing a gradual noising process. At each denoising step, a U-Net model $\epsilon_\theta$ is employed to predict the added noise. This prediction is then used to iteratively reconstruct a clean sample from pure noise, conditioned on both style and text, represented by the embeddings $\boldsymbol{c}_S$ and $\boldsymbol{c}_T$, respectively.

Our U-Net backbone architecture follows that of DiffusionPen and WordStylist, on top of which we integrate classifier-free guidance for two conditions (Brooks et al., 2023). The text condition is fed into the cross-attention layers of the U-Net, while the style embedding is concatenated with the timestep embedding, which informs the model of the noise level in the input. By incorporating the style embedding in this way, every ResNet block in the U-Net has direct access to it, allowing the style embedding to influence the entire image generation process.

Given a line image encoded by a Variational Autoencoder (VAE) (Rombach et al., 2021) as $\boldsymbol{z}_t$, along with the two conditions $\boldsymbol{c}_S$ and $\boldsymbol{c}_T$, the diffusion model is trained to predict the Gaussian noise $\epsilon$ added at each timestep during the forward diffusion process:

$$L = ||\epsilon - \epsilon_\theta(\boldsymbol{z}_t, \boldsymbol{c}_S, \boldsymbol{c}_T)|| \tag{4}$$

To implement the classifier-free guidance, our model should support both conditional and unconditional denoising with respect to the two conditions. To implement this, we separately cancel text and style conditions for 5% of the training instances. Once the model is trained, we sample new synthetic images using the following adaptation of the noise predicted by the U-Net model $\epsilon_\theta$:

$$\epsilon_\theta(\boldsymbol{z}_t, \boldsymbol{c}_S, \boldsymbol{c}_T) = \epsilon_\theta(\boldsymbol{z}_t, \emptyset, \emptyset) + s_T \cdot (\epsilon_\theta(\boldsymbol{z}_t, \boldsymbol{c}_T, \emptyset) - \epsilon_\theta(\boldsymbol{z}_t, \emptyset, \emptyset)) + s_S \cdot (\epsilon_\theta(\boldsymbol{z}_t, \boldsymbol{c}_T, \boldsymbol{c}_S) - \epsilon_\theta(\boldsymbol{z}_t, \boldsymbol{c}_T, \emptyset))$$

$$(5)$$

where $s_T$ and $s_S$ are guidance scales for the text and style conditions. Sampling process illustrated in Eq. 5 assumes that the synthetic images should first be generated using the text condition and then adapted to the handwriting style indicated by the style condition. In this way, we give higher priority to the text condition than to the style constraint. The hyperparameters $s_T$ and $s_S$ are selected by maximizing the cosine similarity between the style embeddings of the generated images and the style embeddings of 20 randomly selected images displaying handwriting from a specific writer.

**Style encoder**   To implement the style encoder, we use the MobileNetV2 model (Sandler et al., 2018). Following a similar approach to (Nikolaidou et al., 2024), we train MobileNetV2 to capture the stylistic characteristics of handwriting by contrastively learning the differences between handwriting styles. The style encoder of DiffusionPen is trained using a triplet loss, which is sensitive to the selection of negative samples (i.e., line images written by different authors with a different handwriting style). Since negative samples can vary significantly in their similarity to the anchor line image, the triplet loss may lead to an inconsistent training process, making it difficult to establish a standardized contrastive learning framework.

To address the above issue, we adopt a softmax-based contrastive loss (Chen et al., 2020), which encourages higher similarity between an anchor line image and a randomly selected positive line image, i.e., one written in the same handwriting style and included in the same batch. The similarity score is normalized over the similarity scores between the anchor and all other line images in the batch. By normalizing across the entire batch, this approach eliminates the need for explicit hard negative mining, as a sufficiently large batch is expected to contain negative samples of varying difficulty.

Knowing that MobileNetV2 encodes the anchor line image $x$ into $f_x$, $f_{pos}$ is a positive image for the anchor image $x$, the batch size is $N$ and $sim(*)$ stands for cosine similarity, we define the softmax-based contrastive loss as follows:

$$L_{contrastive}(f_x, f_{pos}) = \frac{exp(sim(f_x, f_{pos}))}{\sum_{k=1}^{N} exp(sim(f_x, f_k))}$$

$$(6)$$

While the writing particularities are important to allocate a line image to a manuscript, we also need to preserve a certain level of generalization that is required to recognize letters regardless of the writer. For example, while different people write the letter "a" differently, any reader should still be able to recognize the letter. As the contrastive loss might be inclined to group positive embeddings into tight clusters, which might affect this generalization ability of the style encoder, a Sinkhorn-based loss (Sepanj & Fiegth, 2025) is defined to regularize the loss $L_{contrastive}$.

Given $\boldsymbol{S} \in \mathbb{R}^{N \times N}$ as the similarity matrix between all line images of a certain batch, $\boldsymbol{T}$ as a matrix that indicates the transport plan obtained by applying the Sinkhorn–Knopp algorithm (Cuturi, 2013) on $exp(\boldsymbol{S})$ and $\boldsymbol{U} \in \mathbb{R}^{N \times N}$ as a uniform matrix where $U_{i,j} = 1/N$, we compute the Sinkhorn-based loss using the Kullback–Leibler divergence between $\boldsymbol{T}$ and $\boldsymbol{U}$ scaled by the weight $\lambda_{Sinkhorn}$. The final contrastive loss $L_{contrastive}$ is defined as follows:

$$L_{contrastive}(f_x, f_{pos}, T, U) = \frac{exp(sim(f_x, f_{pos}))}{\sum_{k=1}^{N} exp(sim(f_x, f_k))} + \lambda_{Sinkhorn} D_{\mathrm{KL}}(\boldsymbol{T}||\boldsymbol{U})$$

$$(7)$$

# 4 EXPERIMENTS

## 4.1 EXPERIMENTAL SETUP

**Data** To evaluate AdapterTrOCR, we use a collection of five Latin manuscripts written between the 16th and 18th centuries by five different writers. Four of these manuscripts come from our internal collection, with transcriptions prepared by colleagues skilled in Latin and paleography. The fifth manuscript is the *Lateinische Gedichte* volume by Rudolf Gwalther (Stotz & Ströbel, 2021). The number of line transcriptions per manuscript ranges from 866 to 4037. The complete dataset contains about 10K line images, with each transcription averaging 8 tokens and 58.14 characters per line. The full dataset will be released upon acceptance. For the DiffLine training, we exclude the well-known Bullinger dataset (Hodel et al., 2023) because it does not include information about the writers. For the training of AdapterTrOCR, we also exclude the Bullinger dataset as additional training data since it does not provide significant improvements, probably due to the difference in layout and style when compared with our data. More details can be found in Appendix A.5.

**Metrics** To evaluate the HTR task we rely on the HTR-specific metrics like character error rate (CER) and word error rate (WER). Additionally, we include accuracy (Acc) to measure the ability of the models to generate a transcription identical to the ground truth.

**Models for data augmentation with handwritten text generation** We compare our model, Diff-Line for HTG with the following diffusion-based baselines: One-DM (Dai et al., 2024), Diffusion-Pen (Nikolaidou et al., 2024) and WordStylist (Nikolaidou et al., 2023)[4] Additional baseline models that we consider are VATr (Pippi et al., 2023) and HWT (Bhunia et al., 2021). To ensure a fair comparison, all models are fine-tuned on the same dataset.

**Models for line-level HTR** As our model is built on top of TrOCR, which was a state-of-the-art line-level HTR model in 2023, we consider as baselines only models proposed after that year with publicly available code. Therefore, besides TrOCR, we include HTR-VT (Li et al., 2025) and ViTLP (Mao et al., 2024) as additional baselines. We also report results for PyLaia (Puigcerver, 2017), which is the underlying model used for line-level HTR in Transkribus[5], a widely used platform in the digital humanities community. To guarantee a fair comparison, we fine-tune all models on the same dataset. While HTR-VT and PyLaia are line-level HTR models, ViTLP is an end-to-end model. To enable a fair comparison with ViTLP, we run our proposed AdapterTrOCR only on the lines detected by a fine-tuned YOLO model, considering only those with a confidence score above 70%. More details on the line detection process are provided in Appendix A.6.

**Implementation details** To evaluate the effect of the number of annotated line images at the manuscript level, we work with two scenarios. The first scenario evaluates HTR for the manuscript with the smallest number of transcriptions (866), while the second scenario evaluates HTR for the manuscript with the largest number of transcriptions (4037).

We begin by training DiffLine and the HTG baselines on our data collection to enable manuscript-specific data augmentation. Since DiffLine outperforms the baselines (see Section 4.2), we use it to generate 2000 synthetic line images reflecting the handwriting style of the manuscript of each scenario. In total, we generate two sets of 2000 line images, one for each scenario. We use the same arbitrary Latin text to generate the synthetic images of the two scenarios. Further details on the choice to use 2000 synthetic line images per scenario are provided in Appendix A.7.

For each scenario, we define the test set by randomly selecting 300 ground-truth instances from the corresponding manuscript. The training data for AdapterTrOCR in a given scenario consists of: the remaining instances from that manuscript, 2000 synthetic instances generated to capture its handwriting style, and the ground-truth instances from the other four manuscripts. As mentioned above, the 2000 synthetic images are also used to augment the manuscript-specific instances when training the style adapters. Further details on the training procedure and hyperparameter selection for DiffLine and AdapterTrOCR are provided in Appendix A.8.

---

[4]Another recently proposed HTG model is Emuru (Pippi et al., 2025). We do not use this model as a baseline due to the lack of the complete code for training/inference.

[5]https://www.transkribus.org/

Table 1: Comparison between DiffLine and the baseline methods for HTG-based data augmentation (DA) in terms of HTR performance. The HTR results were generated using TrOCR, fine-tuned on the Latin training data specified in the Setup column. GT and Syn- indicate the inclusion of ground-truth and synthetic training data, respectively, for the manuscript associated with each scenario. The DA Method refers to the method used to generate the synthetic data. While CER and WER are standard metrics for evaluating HTR performance, Acc represents the accuracy, measured as the percentage of transcriptions that are counted correct only when the entire line exactly matches the ground-truth text.

| | | 1st Scenario - underrepresented manuscripts | | | 2nd Scenario - well-represented manuscript | | |
|---|---|---|---|---|---|---|---|
| Setup | DA Method | $Acc(\uparrow)$ | $CER(\downarrow)$ | $WER(\downarrow)$ | $Acc(\uparrow)$ | $CER(\downarrow)$ | $WER(\downarrow)$ |
| No GT - No Syn | - | **2.33** | 46.86 | 91.17 | 0.33 | 17.54 | 54.05 |
| No GT - Yes Syn | HWT | 1.23 | 63.23 | 98.86 | 2.12 | 19.19 | 51.55 |
| No GT - Yes Syn | VATr | 1.42 | 61.87 | 97.43 | 3.32 | 19.48 | 50.45 |
| No GT - Yes Syn | One-DM | 1.87 | 55.43 | 93.98 | 6.35 | 17.97 | 47.75 |
| No GT - Yes Syn | WordStylist | 1.66 | 58.09 | 96.32 | 2.66 | 18.06 | 47.68 |
| No GT - Yes Syn | DiffusionPen | 2.00 | 49.46 | 92.91 | 8.67 | 16.03 | 49.87 |
| No GT - Yes Syn | DiffLine | 1.67 | **46.57** | **89.27** | **12.04** | **11.44** | **31.86** |
| Yes GT - No Syn | - | **5.00** | 23.33 | 60.73 | 34.11 | 4.74 | 14.50 |
| Yes GT - Yes Syn | HWT | 4.12 | 25.45 | 62.45 | 34.52 | 5.12 | 14.98 |
| Yes GT - Yes Syn | VATr | 4.23 | 25.04 | 62.47 | 34.45 | 4.65 | 14.98 |
| Yes GT - Yes Syn | One-DM | 4.89 | 23.82 | 61.34 | 36.78 | 4.47 | 14.65 |
| Yes GT - Yes Syn | WordStylist | **5.00** | 24.06 | 61.58 | 36.12 | **4.07** | 14.14 |
| Yes GT - Yes Syn | DiffusionPen | 4.00 | 23.72 | 61.27 | 35.45 | 4.34 | 14.19 |
| Yes GT - Yes Syn | DiffLine | 4.33 | **23.32** | **60.03** | **37.79** | 4.47 | **13.59** |

## 4.2 RESULTS

**Data augmentation with Handwritten Text Generation** We begin by comparing DiffLine with the baseline data augmentation methods in the context of historical Latin HTR. In addition to the scenario where synthetic data is used to extend the training set, we also evaluate a setting in which, for a given manuscript, only synthetic data is used, without any ground-truth annotations. This latter evaluation is important for assessing the standalone quality of the synthetic data. As for this phase, we only want to select the best method for data augmentation, the evaluation is done using only TrOCR fine-tuned on our Latin collection without any task, language and style adaptation. By comparing the proposed DiffLine approach with the baselines (Table1), we observe that, overall, DiffLine produces the most effective synthetic images for improving HTR performance. In contrast, all other HTG baselines appear to degrade performance, increasing both CER and WER. A few synthetic image lines generated by DiffLine and the baselines are presented in Appendix A.9.

Interestingly, Table 1 also shows that generating high-quality synthetic data capable of significantly reducing CER and WER requires a sufficient amount of annotated data for the given manuscript, an observation that reduces the usefulness of synthetic data. In the second scenario, which corresponds to a high-resource manuscript, we observe that the synthetic line images generated by DiffLine are of high quality. When used alone (i.e., without the manuscript-specific real data), they reduce the CER by 34.77% and the WER by 41.05%. However, when both synthetic and real data are used together, the improvements remain significant but are smaller, 5.69% for CER and 6.27% for WER.

On the other hand, when targeting a manuscript with limited training data (first scenario), DiffLine has less information to learn from, resulting in lower-quality synthetic images and smaller reductions in CER and WER. Specifically, when both the real and synthetic data are used to train TrOCR, CER is reduced by only 0.04% and WER by 1.15%. However, for all other HTG baselines in this low-resource setting, both CER and WER increase.

**Handwritten text recognition** To evaluate AdapterTrOCR in each scenario, we train the model on datasets augmented with synthetic data generated by DiffLine, using the handwriting style of the manuscript associated with the respective scenario. The results reported in Table 2 confirm that the proposed AdapterTrOCR outperforms all other HTR baselines in both line-level and end-to-end HTR tasks. Notably, TrOCR consistently ranks as the next best-performing model after AdapterTrOCR, supporting our decision to adopt it as the foundation for our modular adaptation.

In Table 3, we observe that incorporating both historical Latin and style adaptations increases accuracy by 7.62–25.66%, while reducing CER by 13.29-31.76% and WER by 7.49–22.88%. Among

Table 2: Comparison between the proposed AdapterTrOCR and the line-level HTR models TrOCR, HTR-VT and PyLaia. For the comparison with the end-to-end HTR model ViTLP, AdapterTrOCR is applied only to the lines detected by the fine-tuned YOLO model with a confidence score greater than 70% (see Appendix A.6). If a line is not detected by YOLO, its transcription is treated as an empty prediction.

| | 1st Scenario - underrepresented manuscripts | | | 2nd Scenario - well-represented manuscript | | |
|---|---|---|---|---|---|---|
| Method | $Acc(\uparrow)$ | $CER(\downarrow)$ | $WER(\downarrow)$ | $Acc(\uparrow)$ | $CER(\downarrow)$ | $WER(\downarrow)$ |
| ViTLP | 3.53 | 27.43 | 64.63 | **40.02** | 6.22 | 17.43 |
| AdapterTrOCR | **3.89** | **25.53** | **61.35** | 39.54 | **5.34** | **15.42** |
| HTR-VT | 3.87 | 25.35 | 65.25 | 33.63 | 10.53 | 17.46 |
| PyLaia | 1.34 | 29.40 | 67.54 | 26.34 | 12.42 | 20.07 |
| TrOCR | 4.33 | 23.32 | 60.03 | 37.79 | 4.47 | 13.59 |
| AdapterTrOCR | **4.66** | **20.22** | **55.53** | **47.49** | **3.05** | **10.48** |

Table 3: Ablation results for AdapterTrOCR based on the style and historical Latin adaptations. All models are trained on the training data augmented with the synthetic data of the associated scenario.

| | 1st Scenario - underrepresented manuscript | | | 2nd Scenario - well-represented manuscript | | |
|---|---|---|---|---|---|---|
| Method | $Acc(\uparrow)$ | $CER(\downarrow)$ | $WER(\downarrow)$ | $Acc(\uparrow)$ | $CER(\downarrow)$ | $WER(\downarrow)$ |
| TrOCR | 4.33 | 23.32 | 60.03 | 37.79 | 4.47 | 13.59 |
| TrOCR + Latin CLM adapter | 3.66 | 23.67 | 60.12 | 38.00 | 4.03 | 13.55 |
| TrOCR + historical Dutch HTR + Latin CLM adapter | 4.00 | 23.12 | 60.45 | 38.53 | 4.03 | 13.40 |
| TrOCR + historical Latin adapter (Ours) | 4.00 | 22.73 | 60.34 | 40.80 | 4.15 | 12.88 |
| (historical Dutch HTR + (Latin CLM adapter - Dutch CLM adapter)) | | | | | | |
| TrOCR + style adapter without synthetic data | 4.00 | 22.99 | 59.12 | **48.55** | 3.16 | 10.55 |
| TrOCR + style adapter with synthetic data (Ours) | 4.00 | 20.78 | 57.71 | 48.34 | 3.15 | 10.67 |
| AdapterTrOCR (style and historical Latin adapters) | **4.66** | **20.22** | **55.53** | 47.49 | **3.05** | **10.48** |

the two, adapting to the handwriting style of the manuscript proves to be the most significant factor for achieving strong HTR performance. In the second scenario (rich in ground-truth data), augmentation of the data used to train the style adapter generates minimal gains. However, in the first scenario, where the manuscript has limited annotations, the use of synthetic data becomes essential, reducing CER and WER by 6.77% and 2.38%, respectively. Regarding the historical Latin adaptation, we notice that explicitly modeling the language difference (Latin - Dutch) yields better performance than directly adding the Latin adapter (as shown by *TrOCR + historical Dutch HTR + Latin CLM adapter*) because it forces the model to "unlearn" Dutch-like features from the historical HTR adapter. Similarly, removing the historical HTR adaptation in the proxy language (Dutch) (TrOCR + Latin CLM adapter) negatively affects performance, as the model does not aquire the "task knowledge" provided by the historical Dutch HTR adaptation.

## 5 CONCLUSION, LIMITATIONS AND FUTURE WORK

**Conclusions.** In this work, we presented AdapterTrOCR, a modular extension of the TrOCR architecture for recognizing historical Latin handwritten texts. The model introduces two adapter modules: one for historical language adaptation and another for handwriting style adaptation. This design enables effective transfer from a modern English HTR model to historical Latin manuscripts. To mitigate the scarcity of annotated data, we complemented this approach with an HTG model for producing synthetic line images that mimic manuscript-specific handwriting styles. Together, modular adaptation and synthetic data substantially improved recognition accuracy, particularly for underrepresented manuscripts. Moreover, the proposed framework is flexible and can be extended to other languages, making it a valuable tool for cultural heritage preservation.

**Limitations and future work.** As training data plays an essential role in generating accurate HTR results, a straightforward research direction is to improve the quality of synthetic images. Our DiffLine model enhances recognition performance without polluting the training data, but it still struggles with displaying the text correctly. Given this, a promising direction for future work is to address the degradation in text accuracy observed in synthetic line images, where the left side is generally more accurate than the right, likely due to weaker alignment between characters and pixels as the text progresses.

## 6 ETHICS STATEMENT

This work develops an HTR model for recognizing Latin text in digitized historical documents from the 16th–18th centuries. Our data does not contain personal or sensitive information about living individuals, and follows the terms set by the holding institutions. This research aims to support the preservation of cultural heritage and improve scholarly access, with no expected harmful use cases.

## 7 REPRODUCIBILITY STATEMENT

Implementation details of our models and experiments are described in Sections 4.1, with further information about the training setup in the Appendix A.7. Upon acceptance, we will release the full source code, trained models, and datasets used in our experiments to ensure reproducibility.

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
