# A APPENDIX

## A.1 SELECTION OF THE HTR BACKBONE MODEL

Since our training HTR data is limited, we select as backbone model TrOCR a visual-language model already pre-trained for modern English HTR that can effectively learn handwriting patterns for historical Latin HTR. To support our decision, we compare TrOCR with other recent visual-language models like Qwen2 (Yang et al., 2025)[6] and Gemma (Team, 2024).

To run this evaluation, we fine-tune TrOCR and all other models for the second scenario, which focuses on a manuscript that is well represented in the training data. Although Qwen2 and Gemma have more complex architectures than TrOCR, Table 4 shows that TrOCR generally outperforms them. This indicates that TrOCR's HTR-specific pretraining provides a stronger advantage for the Latin HTR task than the larger visual–language models but unfamiliar with HTR.

Table 4: Selection of the HTR backbone: comparison of the HTR-pretrained TrOCR model with larger visual–language models such as Qwen2 and Gemma.

| Method | 2nd Scenario - well-represented manuscript | | |
|--------|----------|----------|----------|
| | $Acc(\uparrow)$ | $CER(\downarrow)$ | $WER(\downarrow)$ |
| Qwen2 | 37.24 | **4.23** | 14.11 |
| Gemma | 35.87 | 6.63 | 16.43 |
| TrOCR | **37.79** | 4.47 | **13.59** |

## A.2 EVALUATION OF ADAPTERTROCR ON NON-LATIN LANGUAGES

While AdapterTrOCR was developed primarily to transcribe old Latin manuscripts, due to the specifics of our collection, it can be easily applied to other languages. We demonstrate this by using a collection of French manuscripts written between the 16th and 18th centuries, totaling 9894 line images, and a collection of German manuscripts from the 16th century, with 8000 line images. The French collection was obtained by merging three CREMMA (Corpus for Recognition and Edition of Medieval and early Modern Archives) corpora from the 16th[7], 17th[8], and 18th[9] centuries with the HTRomance corpus[10], which contains manuscripts written between the 17th and 18th centuries. The German manuscripts belong to the StABS Ratsbücher O10 and Urfehdenbuch X dataset[11].

Since none of these manuscripts provide information about their writers, we cannot distinguish between writing styles. Consequently, AdapterTrOCR does not include the style adaptation, and the training data is not augmented with synthetic samples that require writer-specific information. Therefore, AdapterTrOCR relies solely on the historical language component, which utilises the already trained historical Dutch HTR and the Dutch CLM adapters, as well as a new language CLM adapter for French or German. As shown in Table 5, AdapterTrOCR consistently outperforms the fine-tuned TrOCR. The relatively small margin is due to the absence of the style adaptation, which is the most effective component of AdapterTrOCR (see Table 3).

## A.3 THE EFFECT OF FREEZING THE ENCODER OF TROCR DURING FINE-TUNING

While the TrOCR backbone already benefits from HTR-specific pre-training (Section A.1), the visual domain of its pre-training data can differ from that of our historical manuscripts. TrOCR was pre-trained on modern HTR datasets, whereas our data consist of old documents that show degradation (faded ink, bleed-through, paper texture), different stroke shapes influenced by historical writing tools, and different characteristic ligatures (e.g., ct, st, fi).

---

[6]Despite the availability of the more recent Qwen3 models, we use Qwen2 because a Qwen2 variant is already pretrained for OCR, a task similar to HTR.

[7]https://github.com/HTR-United/CREMMA-MSS-16

[8]https://github.com/HTR-United/CREMMA-MSS-17

[9]https://github.com/HTR-United/CREMMA-MSS-18

[10]https://github.com/HTRomance-Project/modern-roman-languages

[11]https://zenodo.org/records/5153263

Table 5: Comparison between TrOCR and AdapterTrOCR fine-tuned on the French and German HTR datasets. Since these datasets do not include information about the manuscript writers, we cannot apply the writing-style adaptation, and the training data is not augmented with synthetic samples that require writer-specific information. Consequently, AdapterTrOCR relies solely on the historical language adaptation (referred to in this paper as the historical Latin adaptation, due to the specific language of our collection).

| | French Dataset | | | German Dataset | | |
|---|---|---|---|---|---|---|
| Method | $Acc(\uparrow)$ | $CER(\downarrow)$ | $WER(\downarrow)$ | $Acc(\uparrow)$ | $CER(\downarrow)$ | $WER(\downarrow)$ |
| TrOCR | **10.53** | 18.73 | 47.64 | 8.73 | 25.64 | 49.75 |
| AdapterTrOCR (historical language adapter) | 10.48 | **16.98** | **46.89** | 9.03 | **24.12** | **49.09** |

These visual differences are demonstrated in Table 6, where we compare for the second scenario (well-represented manuscript) a fully fine-tuned TrOCR and a variant where only the decoder is fine-tuned while the encoder remains frozen. Because the frozen encoder cannot adapt to the new visual domain, this model is outperformed by the fully fine-tuned version.

Table 6: Comparison between fine-tuned TrOCR and a TrOCR model where only the decoder is fine-tuned while the encoder is frozen.

| | 2nd Scenario - well-represented manuscript | | |
|---|---|---|---|
| Method | $Acc(\uparrow)$ | $CER(\downarrow)$ | $WER(\downarrow)$ |
| TrOCR (frozen encoder) | 34.56 | 6.89 | 15.84 |
| TrOCR | **37.79** | **4.47** | **13.59** |

## A.4 Selection of the Proxy Language for the Historical HTR adaptation

Selecting an appropriate proxy language for the historical HTR adapter should depend not only on linguistic similarity but also on the size and quality of the available training data. Although languages more closely related to Latin than Dutch exist, we chose Dutch because, to the best of our knowledge, the VOC and notarial deeds dataset in Dutch provide the largest human-annotated HTR corpus from the 16th–18th centuries, the same time period as our Latin manuscript collection.

To analyze the impact of linguistic similarity, dataset size, and annotation quality, we also consider the previously introduced German and French datasets (Section A.2). These two languages were selected because they provide relatively large HTR datasets from the same historical period as our collection. To use French or German as proxy languages, we train a new historical HTR adapter on the corresponding dataset and a new CLM-based language adapter in the same language. The rest of the adaptation pipeline remains unchanged. Using the second scenario (well-represented manuscript), Table 7 shows that, despite the linguistic similarity between French and Latin, the size and quality of the Dutch VOC and notarial deeds dataset make Dutch the most effective proxy language for our task.

Table 7: Comparison of the influence of linguistic similarity versus dataset size and annotation quality on the historical HTR adapter. While the Dutch dataset contains more than 40k human-annotated line images, the German and French datasets are much smaller and rely largely on machine-generated transcriptions. Despite the strong linguistic similarity between French and Latin, AdapterTrOCR trained with the Dutch HTR dataset outperforms models whose historical HTR adapter is trained on smaller and less reliable datasets. Results are reported for the second scenario (well-represented manuscripts).

| | 2nd Scenario - well-represented manuscript | | |
|---|---|---|---|
| Method | $Acc(\uparrow)$ | $CER(\downarrow)$ | $WER(\downarrow)$ |
| AdapterTrOCR: German | 48.72 | 3.77 | 11.32 |
| AdapterTrOCR: French | 47.93 | 3.45 | 11.02 |
| AdapterTrOCR: Dutch (Our) | **47.49** | **3.05** | **10.48** |

## A.5 INCLUSION OF THE BULLINGER DATASET IN THE TRAINING DATA

We investigate the impact of including the Bullinger dataset in the training of AdapterTrOCR. We do not consider it for DiffLine training, as the Bullinger data contains multiple handwriting styles (i.e., different writers), and the line images lack annotations specifying the authorship. To evaluate its effect on AdapterTrOCR, we extend the training collection of five manuscripts described in the section dedicated to the experimental setup by adding the Bullinger dataset. We focus on the second scenario, in which the target manuscript is well represented in the training data. In this setup, the dataset is further augmented with synthetic instances generated to match the handwriting style of the target manuscript. As shown in Table 8, adding the well-known Bullinger dataset does not improve the performance of AdapterTrOCR.

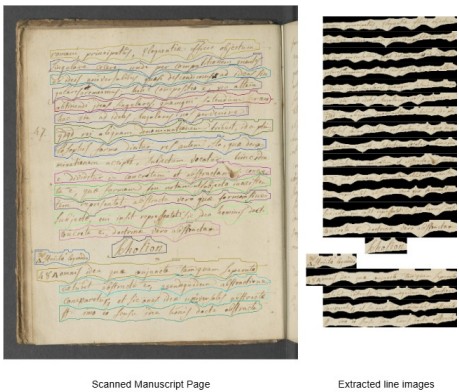

Scanned Manuscript Page          Extracted line images

Figure 2: The scanned manuscript pages (left) are segmented into line images (right).

Table 8: Evaluation of the inclusion of the Bullinger dataset in the training of AdapterTrOCR

| Method | 2nd Scenario - well-represented manuscript | | |
| --- | --- | --- | --- |
| | $Acc(\uparrow)$ | $CER(\downarrow)$ | $WER(\downarrow)$ |
| AdapterTrOCR (with Bullinger) | 46.45 | 3.12 | **10.44** |
| AdapterTrOCR | **47.49** | **3.05** | 10.48 |

## A.6 LINE DETECTION

The task of line detection involves segmenting the scanned manuscript pages into polygons that contain the writing lines. To perform this task, we rely on the most recent version of the YOLO model (Redmon et al., 2016; Khanam & Hussain, 2024), trained on the COCO dataset (Lin et al., 2014), which includes 80 object classes. However, since the class "line" is not part of the COCO class taxonomy, the pre-trained YOLOv11 model cannot segment lines in an image. Therefore, we fine-tune YOLOv11 on the *VOC and notarial deeds dataset* (version 8.1) (Keijser, 2024), a large dataset provided by the National Archives of the Netherlands and the Noord-Hollands Archief. This dataset contains documents from the Dutch East India Company (VOC), written in the 17[th] and 18[th] centuries. After filtering out images that do not contain any lines, we obtain a dataset of 4442 images with annotated polygons, which we use to fine-tune YOLOv11. By setting a 90-10 partition for the training and validation sets, the fine-tuned YOLOv11 model achieves strong results, with over 90% mAP50.

## A.7 SELECTION OF THE NUMBER OF SYNTHETIC INSTANCES

We evaluate the HTR performance of AdapterTrOCR by augmenting the training data with 1000, 1500, 2000, and 2500 synthetic instances, focusing on the second scenario. As shown in Table 9, the results indicate that using 2000 synthetic instances yields the best improvement in performance without introducing noise into the training data.

Table 9: Selection of the number of synthetic data to introduce in the training of AdapterTrOCR.

| | 2nd Scenario - well-represented manuscripts | | |
|---|---|---|---|
| Method | $Acc(\uparrow)$ | $CER(\downarrow)$ | $WER(\downarrow)$ |
| AdapterTrOCR (1000) | 44.61 | 3.54 | 11.12 |
| AdapterTrOCR (1500) | 45.63 | 3.58 | **10.38** |
| AdapterTrOCR (2000) | **47.49** | **3.05** | 10.48 |
| AdapterTrOCR (2500) | 47.01 | 3.22 | 10.49 |

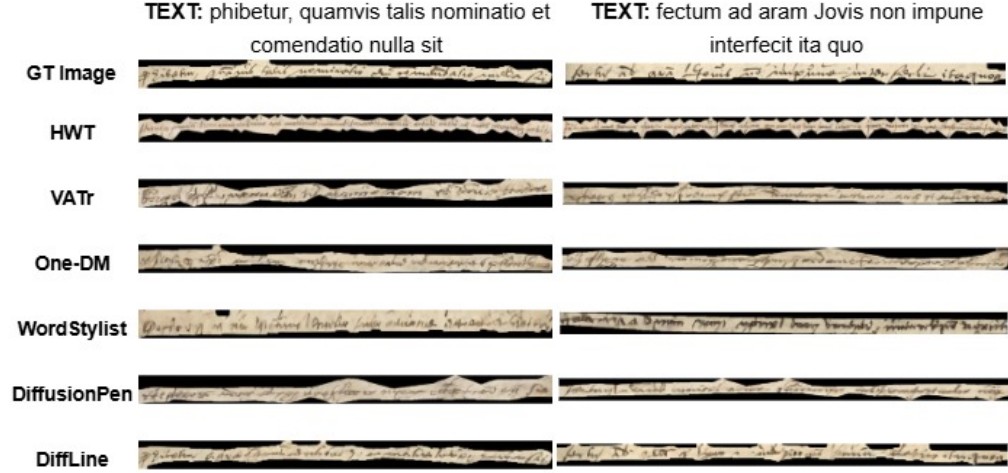

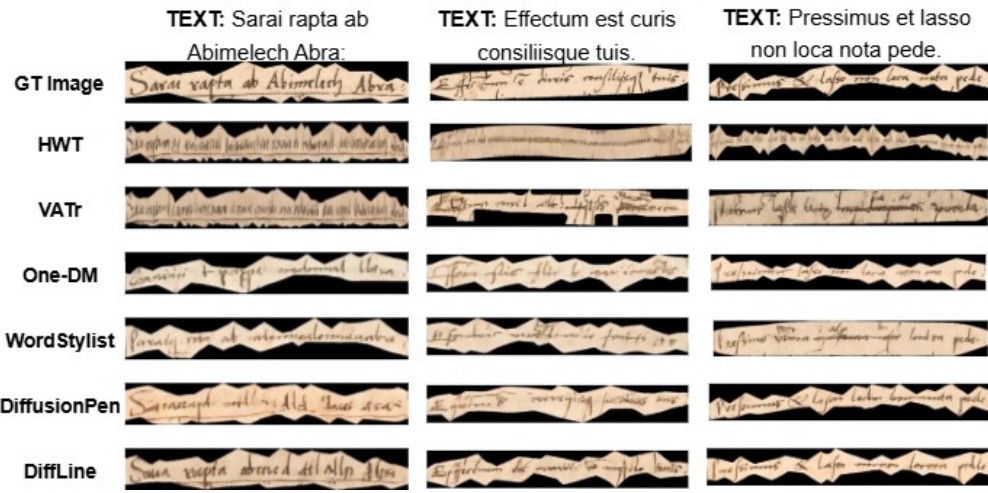

Figure 3: Line images generated by DiffLine and the baselines for the first scenario, which corresponds to an underrepresented manuscript, and for the second scenario, associated with a well-represented manuscript.

## A.8 DIFFLINE AND ADAPTERTROCR TRAINING SETUP

To train DiffLine, we use 1000 epochs, with the line image size set to $64 \times 768$ and a batch size of 64. For the training of the AdapterTrOCR model and its adapters, we set the batch size to 8 and the number of epochs to 20. For the LoRA implementation, the rank $r$ is set to 8, the scaling factor applied to the low-rank update $\alpha_{LoRA}$ is 16, and we apply the LoRA adaptations at the level of both the attention and feed-forward layers. To train the task adapter for historical HTR, we

rely on the Dutch *VOC and notarial deeds dataset* (Keijser, 2024). The Latin and Dutch language adapters are trained using causal language modeling on Wikipedia dumps. The style adapter learns the handwriting style of a manuscript by training on all transcriptions of that manuscript, along with 2000 synthetic line images generated using DiffLine to reflect the handwriting style of the corresponding manuscript. The hyperparameters $\lambda_s$ and $\lambda_l$ are set to 0.3, the hyperparameter $\lambda_{hl}$ is set to 1 and the hyperparameter $\lambda_{Sinkhorn}$ is set to 0.85. Their selection is discussed below.

**Selection of the hyperparameters $\lambda_{hl}$ and $\lambda_s$.** For $\lambda_s$, we test values in $\{0.6, 0.8, 1\}$, and for $\lambda_{hl}$, values in $\{0.1, 0.3, 0.5\}$. For the hyperparameter $\lambda_l$ we select the arbitrary value 0.5. The evaluation is carried out under the second scenario associated to a well-represented manuscript. As shown in Table 10, the optimal values are $\lambda_s = 1$ and $\lambda_{hl} = 0.3$. These results suggest that incorporating handwriting style through the style adapter has a stronger impact on AdapterTrOCR performance than the historical Latin adaptation.

**Selection of the hyperparameters $\lambda_l$.** Considering the optimal values for the hyperparameters $\lambda_{hl}$ and $\lambda_s$, we determine the optimal value of the hyperparameter $\lambda_l$ by choosing between the values $\{0.1, 0.3, 0.5\}$. According to Table 11, the optimal value is 0.3.

All experiments were run on NVIDIA GeForce RTX 3090 GPUs. Training DiffLine requires 41 GPU hours. Training AdapterTrOCR takes 29.6 GPU hours, compared to 27.8 GPU hours for TrOCR. AdapterTrOCR additionally uses four adapters, each requiring an average of 5.5 GPU hours to train; however, these adapters can be reused when training new AdapterTrOCR models.

Table 10: Selection of the hyperparameters $\lambda_{hl}$ and $\lambda_s$.

| Method | 2nd Scenario - well-represented manuscript | | |
|---|---|---|---|
| | $Acc(\uparrow)$ | $CER(\downarrow)$ | $WER(\downarrow)$ |
| AdapterTrOCR ($\lambda_s$=0.6, $\lambda_{hl}$ = 0.1) | 46.73 | 3.715 | 10.98 |
| AdapterTrOCR ($\lambda_s$=0.6, $\lambda_{hl}$ = 0.3) | 46.86 | 3.72 | 11.02 |
| AdapterTrOCR ($\lambda_s$=0.6, $\lambda_{hl}$ = 0.5) | 46.76 | 3.43 | 11.15 |
| AdapterTrOCR ($\lambda_s$=0.8, $\lambda_{hl}$ = 0.1) | 46.95 | 3.52 | 10.86 |
| AdapterTrOCR ($\lambda_s$=0.8, $\lambda_{hl}$ = 0.3) | 47.09 | 3.41 | 11.14 |
| AdapterTrOCR ($\lambda_s$=0.8, $\lambda_{hl}$ = 0.5) | 47.07 | 3.58 | 11.05 |
| AdapterTrOCR ($\lambda_s$=1, $\lambda_{hl}$ = 0.1) | **47.65** | 3.13 | 10.80 |
| AdapterTrOCR ($\lambda_s$=1, $\lambda_{hl}$ = 0.3) | 47.54 | **3.12** | **10.50** |
| AdapterTrOCR ($\lambda_s$=1, $\lambda_{hl}$ = 0.5) | 47.39 | 3.17 | 10.63 |

Table 11: Selection of the hyperparameter $\lambda_l$.

| Method | 2nd Scenario - well-represented manuscript | | |
|---|---|---|---|
| | $Acc(\uparrow)$ | $CER(\downarrow)$ | $WER(\downarrow)$ |
| AdapterTrOCR ($\lambda_l$=0.1) | 47.58 | 3.07 | 10.51 |
| AdapterTrOCR ($\lambda_l$=0.3) | **47.49** | **3.05** | **10.48** |
| AdapterTrOCR ($\lambda_l$=0.5) | 47.54 | 3.12 | 10.50 |

A.9 HANDWRITTEN TEXT GENERATION: QUALITATIVE ANALYSIS

Figure 3 shows examples of images generated by DiffLine and the HTG baselines for the first scenario, which corresponds to an underrepresented manuscript, and for the second scenario, associated with a well-represented manuscript. As expected, all models perform better in the second scenario. While all baselines struggle to generate accurate text, especially HWT, DiffusionPen is the closest baseline, occasionally producing well-defined characters. Compared to the baselines, DiffLine generates less noisy image lines. However, it still has difficulty properly aligning the generated text with the input text. Interestingly, DiffLine produces more accurate text on the left side of the images, with accuracy decreasing toward the right. We suspect this behavior results from weaker alignment between pixels and characters as the text progresses, possibly due to inconsistent spacing between words in the handwritten training data.