# OpenReview forum: "Handwritten Text Recognition Adaptation for Low-Resource Languages: A Case Study on Historical Latin Manuscripts"
_ICLR.cc/2026/Conference — Submitted to ICLR 2026_

### Official Review · Reviewer_UKgu · 2025-10-20

**Soundness:** 2
**Presentation:** 4
**Contribution:** 2
**Rating:** 4
**Confidence:** 3

**Summary:**

This paper presents AdapterTrOCR, a novel model for Handwritten Text Recognition (HTR) specifically designed for historical Latin manuscripts, a challenging task due to the low-resource nature of the language and the variability of handwriting styles. The proposed model extends the TrOCR architecture by incorporating two distinct adapter modules: one for adapting to the historical Latin language and another for adapting to specific handwriting styles. To address the scarcity of annotated data, the authors also introduce DiffLine, a Handwritten Text Generation (HTG) technique for data augmentation. The experimental results demonstrate the effectiveness of this approach, showing significant reductions in Character Error Rate (CER) and Word Error Rate (WER).

**Strengths:**

1. The proposed AdapterTrOCR model is innovative. The modular design, which separates historical language adaptation from handwriting style adaptation, is a logical and effective way to handle the complexities of historical manuscripts.

2. The paper tackles the common problem of data scarcity in low-resource scenarios by proposing a new data augmentation method, DiffLine. This is a valuable contribution that could be beneficial for other HTR tasks as well.

3. The authors provide a comprehensive evaluation of their model. The reported reductions in CER by 13.33% and WER by 8.56% are substantial.

4. The paper is well-organized and clearly written. The methodology is explained in detail

**Weaknesses:**

1. The paper does not include a comparison with existing Large Language Models (LLMs), which have demonstrated strong performance on various text recognition and understanding tasks. Evaluating AdapterTrOCR against state-of-the-art multimodal LLMs could provide a more complete picture of its capabilities.

2. The study focuses exclusively on historical Latin manuscripts. While the results are promising, the paper would be stronger if it included experiments on other low-resource languages or different types of historical documents to demonstrate the generalizability of the proposed method.

3. The historical language adaptation relies on a proxy language (Dutch). The choice of this proxy language could impact the model's performance. The paper could benefit from a discussion on how sensitive the model is to the choice of the proxy language.

4. The proposed solution involves a multi-step pipeline that includes training adapters, generating synthetic data with a diffusion model, and fine-tuning the final model. This complexity might pose a challenge for researchers and practitioners who are not experts in all of these areas.

**Questions:**

Your work builds a complex, specialized model. Based on my personal experience, however, current large vision–language models (e.g., GPT-4o and Gemini-2.5-Pro) already demonstrate remarkable performance on complex or ambiguous image–text recognition tasks in few-shot and even zero-shot settings. What are the advantages of your proposed method?

---

> ### Author Response · Authors · 2025-11-22
> **Official Comment by Authors to Reviewer UKgu**
>
> We thank the reviewer for the feedback and comments. Our responses to the weaknesses and questions are given below.
>
> **Weaknesses:**
>
> 1.  	The paper does not include a comparison with existing Large Language Models (LLMs), which have demonstrated strong performance on various text recognition and understanding tasks. Evaluating AdapterTrOCR against state-of-the-art multimodal LLMs could provide a more complete picture of its capabilities.
>
> Response: **We compare fine-tuned TrOCR with fine-tuned Qwen2 and Gemma and observe that, despite Qwen2 and Gemma being larger models, they are disadvantaged because, unlike TrOCR, they are not pre-trained for the HTR task. Further details are provided in our response to Reviewer 2uJK and between the lines 151-152 and in Appendix A1. Since TrOCR outperforms both models, it follows that AdapterTrOCR also outperforms them.**
>
> 2.  	The study focuses exclusively on historical Latin manuscripts. While the results are promising, the paper would be stronger if it included experiments on other low-resource languages or different types of historical documents to demonstrate the generalizability of the proposed method.
> Response: **We show that AdapterTrOCR can outperform TrOCR not only for Latin but also for other languages. We prove this on two additional HTR datasets written in German and French. Further details are provided above in our response to Reviewer 2uJK** and between the lines 160-161 and Appendix A2.
>
> 3.     The historical language adaptation relies on a proxy language (Dutch). The choice of this proxy language could impact the model's performance. The paper could benefit from a discussion on how sensitive the model is to the choice of the proxy language
> Response: **We selected Dutch as the proxy language because, according to our knowledge, it provides the largest high-quality, human-annotated HTR dataset from the same historical period as our Latin manuscripts.** Experiments with smaller French and German datasets, whose annotations are mostly machine-generated, show that dataset size and accuracy matter more for HTR  performance than linguistic similarity between languages. The French and German languages were selected because they offer large datasets from the same historical interval as our collection. The results below are presented for the second scenario of our analysis (well-represented manuscript) (lines 205-211 and Appendix A4).
> | Method                     | ACC   | CER  | WER  |
> |----------------------------|-------|------|------|
> | AdapterTrOCR: German       | 48.72 | 3.77 | 11.32|
> | AdapterTrOCR: French       | 47.93 | 3.45 | 11.02|
> | AdapterTrOCR: Dutch (Ours) | **47.49** | **3.05** | **10.48**|
>
> 4.  	The proposed solution involves a multi-step pipeline that includes training adapters, generating synthetic data with a diffusion model, and fine-tuning the final model. This complexity might pose a challenge for researchers and practitioners who are not experts in all of these areas.
> Response: **We acknowledge that our pipeline involves several steps and can look complex, but this structure reflects the end-to-end, production-oriented nature of our task. Each component is modular and can be used independently, allowing researchers to adopt only the parts relevant to their setting. Through our pipeline, our goal is to provide a diverse set of instruments that hopefully will help future researchers to solve HTR tasks in low-resource languages and domains.**
>
> **Questions:**
>
> 1.     Your work builds a complex, specialized model. Based on my personal experience, however, current large vision–language models (e.g., GPT-4o and Gemini-2.5-Pro) already demonstrate remarkable performance on complex or ambiguous image–text recognition tasks in few-shot and even zero-shot settings. What are the advantages of your proposed method?
>
> Response: While  GPT-4o and Gemini-2.5-Pro are not entirely free, we compare fine-tuned TrOCR with fine-tuned Gemma (a lighter model inspired by Gemini) and fine-tuned Qwen2 (as suggested by reviewer 2uJK). **Despite being larger models, both Gemma and Qwen are outperformed by TrOCR, as shown in our response to Reviewer 2uJK. We attribute this to TrOCR’s pre-training on HTR.** This highlights the importance of training data. In low-resource settings such as Latin HTR, our adapter-based model and data augmentation strategy offer a practical and effective way to compensate for limited training data.

---

> > ### Comment · Reviewer_UKgu · 2025-11-22
> >
> > Thank you, you mentioned that:
> > > We compare TrOCR with Qwen2, for which an OCR-pretrained checkpoint is available (we couldn’t find an OCR-based checkpoint for Qwen3), and with Gemma
> >
> > Which specific variants of Gemma and Qwen2 did you use, and what are their parameter sizes? As far as I know, Qwen2 is already a relatively old model whose performance lags far behind more recent systems. From the results you reported, TrOCR does not seem to have a large advantage over Gemma and Qwen2. This makes me think that the zero-shot performance of current frontier models (GPT-5, Gemini-2.5-Pro, Qwen3-VL) is quite likely to already surpass the model trained in your paper.
> >
> > You also mentioned that
> > > couldn’t find an OCR-based checkpoint for Qwen3
> >
> > > GPT-4o and Gemini-2.5-Pro are not entirely free
> >
> > These two reasons are not very convincing to me as arguments against comparing with the strongest available models. Qwen3-VL is a general-purpose VLM and does not require a special OCR-based checkpoint, and using GPT-4o or Gemini-2.5-Pro only requires calling their APIs, the cost of doing so seems much lower than building and training the complex pipeline proposed in this work.
> >
> > Even if your model underperforms systems like Qwen3-VL or GPT-4o or Gemini-2.5-Pro, quantifying the performance gap would still be important and informative.

---

### Official Review · Reviewer_Vj1a · 2025-10-31

**Soundness:** 3
**Presentation:** 3
**Contribution:** 2
**Rating:** 2
**Confidence:** 3

**Summary:**

This paper addresses handwritten text recognition (HTR) for low-resource languages, focusing on historical Latin. The authors propose two main components: (1) TrOCR adaptation through task and language decomposition, and (2) synthetic handwritten data generation. For adaptation, TrOCR is first trained on historical Dutch to capture general HTR ability, and two LoRA adapters (one for Latin and one for Dutch) are then used to extract Latin-specific language knowledge via parameter differencing. To mitigate data scarcity, the authors generate synthetic handwriting using DiffusionPen with dual classifier-free guidance and MobileNetV2 for style encoding. Overall, the paper is well-structured and experimentally thorough, with clear motivation and creative integration of modern HTR and diffusion techniques. However, the core contributions are largely incremental, offering limited performance gains and uncertain justification for the proposed adaptation strategy.

**Strengths:**

+ The paper presents a novel decomposition of HTR into task and language adaptations, using historical Dutch as a proxy for historical-domain adaptation and leveraging cross-language differencing (Latin vs. Dutch) to isolate linguistic knowledge.

+ The experimental setup is comprehensive and well-structured, with evaluations conducted across multiple models (TrOCR, PyLaia, DiffusionPen, and WordStylist)  for both HTR and HTG tasks.

+ The paper provides comprehensive experimental results such as ablation studies, HTG effectiveness, and HTR performance.

+ The work demonstrates strong integration and adaptation of modern HTR and diffusion techniques, showing practical creativity in combining them for low-resource historical languages.

**Weaknesses:**

- Limited performance improvement. The reported gains for both HTR and HTG are modest and, in some cases, inconsistent. For example, in Table 1 for yes GT, yes Synth 1st scenario, DiffLine achieves a negligible 0.01% CER improvement over base TrOCR, while the accuracy of the proposed method decreases by 0.67% compared to base TrOCR.

- Questionable effectiveness of the language adaptation strategy. Theoretically, the language adaptation strategy is interesting and may help but there is no significant experimental evidence. Training a LoRA adapter on historical Dutch and then applying a differential LoRA derived from Latin-Dutch subtraction may not meaningfully differ from directly training a Latin LoRA adapter, raising doubts about the necessity of this decomposition. For example: TrOCR + Dutch + (Latin - Dutch) = TrOCR + Latin

- Incremental novelty. Many core components rely on adapting existing architectures (e.g., TrOCR, DiffusionPen) via LoRA fine-tuning or different existing loss functions. While the integration is thoughtful, the methodological contribution is limited with no significant performance gains.

- Minimal advantage over baseline models. The base TrOCR achieves comparable performance, improving CER from 23.32% to 20.22% after adaptation, suggesting limited practical impact.

- Limited practicality of synthetic data generation. Although the paper emphasizes its low-resource applicability, it acknowledges that HTG quality scales with dataset size, which undermines its practicality for truly low-resource languages.

- Concerns about generalization. The authors note that including the Bullinger dataset does not yield better results due to differences in layout and style. This raises questions about the robustness and generality of the proposed adaptation pipeline across varied handwriting sources.

- While the paper demonstrates solid experimental effort, the core contributions are incremental and the reported gains minimal. The proposed adaptation strategy does not convincingly justify its added complexity or conceptual novelty.

**Questions:**

1. What specific challenges in historical Latin make it distinct from modern English for HTR, beyond style variation? Since both use the same alphabet, could this explain why the base TrOCR performs competitively?

2. Why is style adaptation implemented in the decoder of TrOCR? Wouldn’t the encoder, or a combined adaptation strategy, better capture handwriting features?

3. What motivated the choice of generating 2,000 synthetic line images? Was this empirically optimized, or constrained by computational cost?

4. In Table 3, the style adapter appears to contribute most of the improvement, while the language adapter provides minimal gains, even when combined. Could the authors elaborate on this asymmetry?

---

> ### Author Response · Authors · 2025-11-22
> **Official Comment by Authors to Reviewer Vj1a (First Part)**
>
> We thank the reviewer for the feedback and comments. Our responses to weaknesses 1-5 are given below.
> 1.      Limited performance improvement. The reported gains for both HTR and HTG are modest and, in some cases, inconsistent. For example, in Table 1 for yes GT, yes Synth 1st scenario, DiffLine achieves a negligible 0.01% CER improvement over base TrOCR, while the accuracy of the proposed method decreases by 0.67% compared to base TrOCR
> Response: **Table 1 demonstrates the effectiveness of our HTG method (DiffLine), particularly when compared with recent state-of-the-art HTG approaches: it is the best substitute when no ground truth (GT) is available and the only method that does not degrade training data quality when GT is included (lines 414–423)**. As the reviewer notes, adding synthetic data directly to the HTR model does not yield large performance gains. However, Table 3 shows that synthetic data is important for training the style adapter, helping it to learn handwriting styles when GT for a given style is limited (lines 459–462). **Overall, the improvements of our full method are substantial, reducing CER and WER by up to 46.6% and 29.7% in relative terms, respectively (Table 3).**
>
> 2.      Questionable effectiveness of the language adaptation strategy. Theoretically, the language adaptation strategy is interesting and may help but there is no significant experimental evidence. Training a LoRA adapter on historical Dutch and then applying a differential LoRA derived from Latin-Dutch subtraction may not meaningfully differ from directly training a Latin LoRA adapter, raising doubts about the necessity of this decomposition. For example: TrOCR + Dutch + (Latin - Dutch) = TrOCR + Latin.
> Response: While the reviewer is right to raise this question, **the decomposition “TrOCR + Dutch + (Latin − Dutch)” that is actually, TrOCR + Historical Dutch HTR + (Latin – Dutch), allows us to better isolate the transition from a modern English TrOCR to a historical Latin TrOCR.** Since our Latin HTR training data is limited, **removing the historical HTR** adaptation (even when it is provided through a proxy language such as Dutch) **would negatively impact performance: without it, the model would lack the task-specific knowledge contributed by the historical Dutch HTR adaptation** (lines 465-467 and Table 3).
>
> 3.      Incremental novelty. Many core components rely on adapting existing architectures (e.g., TrOCR, DiffusionPen) via LoRA fine-tuning or different existing loss functions. While the integration is thoughtful, the methodological contribution is limited with no significant performance gains.
> Response: Our **main contribution is the decomposition rule that enables transitioning from a modern English HTR model to one suited for historical Latin HTR by separating task ability (historical HTR in a proxy language), language ability (proxy language − Latin), and style ability.** This framework, validated extensively for Latin, is also applicable to other languages (lines 160-162 and Appendix A2). **In addition, we introduce a new HTG model built on DiffusionPen but enhanced with classifier-free guidance, allowing better control of style and text conditioning and producing higher-quality synthetic images than recent state-of-the-art methods. Together, these contributions yield substantial performance gains, reducing CER and WER by up to 46.6% and 29.7%, and improving accuracy by up to 10.4%.**
> 4.      Minimal advantage over baseline models. The base TrOCR achieves comparable performance, improving CER from 23.32% to 20.22% after adaptation, suggesting limited practical impact.
> Response: While the CER reduction from 23.32% to 20.22% may appear modest in absolute terms, it corresponds to a meaningful improvement when evaluated in relative terms. For the same scenario, an underrepresented manuscript in the training data, we also observe a WER reduction from 60.03% to 55.53%. Moreover, in scenarios where the manuscript is well represented in the training set, the gains are substantial (as responded to the above comment).
>
> 5.      Limited practicality of synthetic data generation. Although the paper emphasizes its low-resource applicability, it acknowledges that HTG quality scales with dataset size, which undermines its practicality for truly low-resource languages.
> Response: **While adding synthetic data directly to the HTR training set has limited impact, it plays an important role in helping the style adapter to learn diverse handwriting styles. These learned styles are then integrated into our adaptation module, significantly improving HTR performance, especially for underrepresented manuscripts. This claim  is demonstrated in Table 3 and discussed in lines 458–462. Given the fact that the quality of HTG scales with dataset size is a general limitation of all HTG models, we believe that our training strategy is meaningful and can help future research to better integrate synthetic data into the HTR models.**

---

> > ### Author Response · Authors · 2025-11-22
> > **Official Comment by Authors to Reviewer Vj1a (Second part)**
> >
> > We thank the reviewer for the feedback and comments. Our responses to weaknesses 6-7 and the questions 1-4 are given below.
> >
> > **Weaknesses (6-7):**
> >
> > 6.      Concerns about generalization. The authors note that including the Bullinger dataset does not yield better results due to differences in layout and style. This raises questions about the robustness and generality of the proposed adaptation pipeline across varied handwriting sources.
> > Reponse: We are afraid there might be a misunderstanding here. **We developed our adaptation system only because our Latin HTR training data were insufficient. For the same reason, we also experimented with adding the Bullinger dataset, a large well-known historical Latin HTR dataset (lines 347–350). However, in our case, Bullinger data did not improve our results** probably because of the difference in layout in style or probably because Bullinger is mostly generated by Transkribus. **We did not evaluate AdapterTrOCR on Bullinger, as our goal is to transcribe our own documents.**
> > To address the concerns related to generalization to other languages, we added an evaluation of our model for two new datasets with historical data written in French and German. More details are given as a response to Reviewer 2uJK and also between the lines 160-161 and in Appendix A2).
> >
> > 7.      While the paper demonstrates solid experimental effort, the core contributions are incremental and the reported gains minimal. The proposed adaptation strategy does not convincingly justify its added complexity or conceptual novelty.
> > Response: **With our paper we aim to provide a complete pipeline for historical Latin HTR,** combining model adaptation and synthetic data generation to address low-resource and diverse handwriting challenges. **The decomposition strategy—separating task ability, language adaptation, and style adaptation—is novel and can be applicable to other languages, and together with our novel HTG module, it delivers substantial practical gains (CER and WER reductions up to 46.6% and 29.7%, accuracy improvements up to 10.4%).**
> >
> > **Questions:**
> > 1.      What specific challenges in historical Latin make it distinct from modern English for HTR, beyond style variation? Since both use the same alphabet, could this explain why the base TrOCR performs competitively?
> > Response: Beyond stylistic variation, historical manuscripts can differ from modern ones in stroke shapes (influenced by writing tools), ligatures (e.g., “ct,” “st,” “fi”), and document degradation (faded ink, ink bleed, paper texture). Many of these visual patterns can be learned from manuscripts of the same period, even in a different language like Dutch. Moreover, since Dutch manuscripts originate from a region geographically close to the source of our Latin collection, similar handwriting and layout patterns are expected. This is why our model adaptation incorporates not only a writing style adapter but also a historical Dutch HTR adaptation.  The reason why base TrOCR is competitive is because it is fine-tuned on the same Latin HTR dataset used to fine-tune AdapterTrOCR (as it is the case for all our baselines). This is step is required for a fair comparison.
> >
> > 2.      Why is style adaptation implemented in the decoder of TrOCR? Wouldn’t the encoder, or a combined adaptation strategy, better capture handwriting features?
> > Response: While the encoder could also benefit from a style adapter, we applied adaptation exclusively at the decoder level because language adaptations can only be implemented there. **To ensure a uniform integration of all adapters, we therefore applied all adaptations solely at the decoder (lines 158–161).**
> >
> > 3.      What motivated the choice of generating 2,000 synthetic line images? Was this empirically optimized, or constrained by computational cost?
> > Response: **The number of synthetic instances is selected empirically to avoid introducing excessive noise into the training data.** This analysis is presented in our paper (lines 366–367), with additional details provided in Appendix A7.
> >
> > 4.      In Table 3, the style adapter appears to contribute most of the improvement, while the language adapter provides minimal gains, even when combined. Could the authors elaborate on this asymmetry?
> > Response: The style adapter has the greatest impact because, after integrating all adapters into the TrOCR architecture, the model is further fine-tuned on a Latin HTR dataset containing manuscripts with various writing styles. **This fine-tuning increases exposure to historical Latin text but provides limited exposure to specific handwriting styles. Therefore, adding a dedicated style adapter is important for capturing and inducing particular writing style characteristics,** enabling accurate recognition of text in that style.

---

> > > ### Comment · Reviewer_Vj1a · 2025-11-27
> > > **Reviewer's comments on rebuttal**
> > >
> > > I thank the authors for their detailed and well-structured rebuttal and appreciate the clarifications regarding the adaptation framework, the synthetic data role, and the additional experiments.
> > >
> > > **(1) Limited performance improvement.**
> > > Thank you for highlighting the relative CER/WER reductions and comparisons against other HTG baselines, but my concern was about absolute gains over a strong TrOCR baseline. In Table 3 (Scenario 1), CER improves from 23.32 to 20.22 and accuracy from 4.33 to 4.66, which remain modest given the pipeline's complexity. This point remains only partially addressed.
> > >
> > > **(2) Effectiveness of the language adaptation strategy.**
> > > While the language decomposition performs slightly better than directly applying a Latin CLM, its absolute improvement over base TrOCR remains small. Most of the benefit comes from the style adapter rather than the language adapter. The decomposition is conceptually motivated, but its empirical contribution remains limited, and its necessity remains unclear.
> > >
> > > **(3) Incremental novelty.**
> > > The decomposition and HTG enhancements are valuable engineering efforts, but they extend existing architectures. The rebuttal does not provide new evidence that changes the incremental nature of the contributions.
> > >
> > > **(4) Minimal advantage over baseline models.**
> > > Thank you for the clarification about the reported improvements, but the gains for the underrepresented scenario remain modest. While larger improvements occur in the well-represented setting, this does not directly address the paper’s low-resource motivation. This concern remains.
> > >
> > > **(5) Practicality of synthetic data generation.**
> > > The role of synthetic data in supporting the style adapter is now clearer. While this explanation is helpful, it does not change the practical limitation. HTG quality still scales with dataset size, and this limits its usefulness in genuinely low-resource settings. The concern is partially addressed but not resolved.
> > >
> > > **(6) Generalization concerns.**
> > > The French and German evaluations are appreciated, but their limited scope and small gains, together with the fact that the Bullinger dataset provides little benefit due to layout/style differences, leave generalization insufficiently demonstrated.
> > >
> > > **(7) Overall incremental contribution and complexity.**
> > > The rebuttal restates the pipeline's goals but does not address the relatively small empirical gains given the added architectural complexity.
> > >
> > > **Questions 1-4.** All clarifications were answered satisfactorily.
> > >
> > > While the rebuttal improves clarity, the primary concerns remain. I therefore maintain my rating.

---

### Official Review · Reviewer_aFsT · 2025-10-31

**Soundness:** 3
**Presentation:** 3
**Contribution:** 3
**Rating:** 4
**Confidence:** 4

**Summary:**

The authors proposed a modular and parameter-efficient approach to adapt TrOCR, a OCR system that is trained on modern English to historical Latin. The idea is to use LoRA adapters on TrOCR decoder to separate the language ability from Dutch to Latin CLM and add a style adapter that can be per-document. This is very interesting for low-resource, writer-specific manuscripts. After that a diffusion-based generator for handwritten text introduces dual classifier-free guidance for text and style. At the end MobileNetV2 style encoder trained with InfoNCE and a Sinkhorn regularizer is used.

I liked the approach of dividing the “historical HTR task ability” and “language ability,” which could also be applied in other tasks. However, I miss some robustness checks to attribute improvements to the proposed modules rather than to stronger decoding or to overlap between training and testing styles.

**Strengths:**

The idea of decomposing adaptation into historical HTR “task ability”, cross-lingual “language ability,” and manuscript-specific style via additive LoRA composition is very interesting, especially for the handwriting recognition context.

The proposed approach improves CER/WER over traditional and strong baselines, which is promising. The proposed ablations show that the style adapter provides the largest lift, with language/task adapters adding further gains. The diffusion module clearly outperforms other generators when GT is ample.

The results show that the approach is practically relevant, and the modular PEFT approach is reproducible and budget-friendly.

**Weaknesses:**

Not sure I understand correctly but the historical Latin adapter is trained with CLM and the final model is “subsequently fine-tuned”. In this case, I don't see the control with a frozen TrOCR plus shallow-fusion Latin LM or LM-KL reranking. Without those, improvements may partially reflect stronger decoding priors rather than adapter composition.

It seems that the train/test splits are within the same manuscript/writer. Because the style adapters are tuned using 20 “randomly selected images” from a specific writer to choose sT and sS. Considering that they could come from the full manuscript pool, the hyperparameter search may peek at test-style statistics. I think the use of leave-one-manuscript-out evaluation is needed.

The authors suggest all adapters are trained on the decoder “as some tasks are language-only.” For the historical HTR adapter, which raises a modality mismatch, if only the decoder is adapted, improvements might come from a better language model rather than better visual grounding.

I think applying YOLO filtering and a 70% confidence cutoff only for AdapterTrOCR in the ViTLP comparison complicates fairness because differences in missed/false lines can dominate CER/WER.

The “single author per manuscript” assumption is fragile for historical collections. If violated, per-manuscript style adapters may entangle writer and content, reducing portability to mixed-hand manuscripts.

**Questions:**

Since adapters are composed additively, have the authors tested whether the order of adapter application (style -> language -> task vs. the reverse) affects convergence or performance?

What are the total training times and GPU hours for AdapterTrOCR and DiffLine? How do they compare with full fine-tuning and prior PEFT baselines?

Could the adapter composition strategy generalize to other low-resource scripts (e.g., Old French, Medieval Spanish)? Have preliminary tests been done?

---

> ### Author Response · Authors · 2025-11-22
> **Official Comment by Authors to Reviewer aFsT (Weaknesses)**
>
> We thank the reviewer for the feedback and comments. Our responses to the weaknesses are given below.
>
> 1.      Not sure I understand correctly but the historical Latin adapter is trained with CLM and the final model is “subsequently fine-tuned”. In this case, I don't see the control with a frozen TrOCR plus shallow-fusion Latin LM or LM-KL reranking. Without those, improvements may partially reflect stronger decoding priors rather than adapter composition.
> Response: In our approach, each adapter is trained independently and then integrated into the TrOCR backbone. After integrating the adapter weights, the resulting model, AdapterTrOCR, is further fine-tuned on the Latin HTR data (lines 188–196). As TrOCR is already pre-trained for HTR; in our preliminary experiments, we fine-tuned only the decoder while freezing the encoder to preserve its visual capabilities. However, **full fine-tuning of TrOCR produced better results, likely because the visual domain of our Latin data differs from that of the pre-training dataset (lines 214-215 and Appendix A3).** The results are provided for the second scenario (well-represented manuscript).
> | Method                 | ACC   | CER   | WER   |
> |------------------------|-------|-------|-------|
> | TrOCR (frozen encoder) | 34.56 | 6.89  | 15.84 |
> | TrOCR                  | **37.79** | **4.47**  | **13.59** |
>
> 2.      It seems that the train/test splits are within the same manuscript/writer. Because the style adapters are tuned using 20 “randomly selected images” from a specific writer to choose sT and sS. Considering that they could come from the full manuscript pool, the hyperparameter search may peek at test-style statistics. I think the use of leave-one-manuscript-out evaluation is needed.
> Response: For clarity, the style adapters are used only for the HTR model, not for the HTG model. The 20 reference images are used solely to compute style similarity for selecting the guidance hyperparameters (sT, sS) of the HTG model. **Since the images generated by the HTG model are conditioned on Latin texts not included in HTG/HTR datasets, no data used further for training/testing HTR is used, and there is no data leakage. Consequently, leave-one-manuscript-out evaluation is not needed for our HTG/HTR procedure.**
>
> 3.      The authors suggest all adapters are trained on the decoder “as some tasks are language-only.” For the historical HTR adapter, which raises a modality mismatch, if only the decoder is adapted, improvements might come from a better language model rather than better visual grounding.
> Response: **Our experiment with a frozen encoder (first answer) shows that improvements do not result from a stronger language model alone.** However, as the reviewer suggests some adaptations should target not only the decoder but also the encoder. Although we considered this option initially, **we chose to place all adapters on the decoder for two reasons: (1) consistency reasons:** the adapters implement the transition from a modern English HTR model to a historical Latin HTR model and should all be applied in the same way, and **(2) the language adapters must “unlearn” the Dutch-like linguistic biases introduced by the historical HTR adapter, which is only possible if all adapters operate at the decoder level (where the language adapters can be applied).**
>
> 4.      I think applying YOLO filtering and a 70% confidence cutoff only for AdapterTrOCR in the ViTLP comparison complicates fairness because differences in missed/false lines can dominate CER/WER.
> Response: When we compare AdapterTroCR with ViTLP, **if a line is not detected by YOLO, its transcription is treated as an empty prediction (lines 436-437). The same procedure is applied consistently when computing CER/WER, ensuring that differences in line detection do not bias the comparison.**
>
> 5.      The “single author per manuscript” assumption is fragile for historical collections. If violated, per-manuscript style adapters may entangle writer and content, reducing portability to mixed-hand manuscripts.
> Response: In our collection, **each manuscript consists of student notes for individual disciplines**, which serve as personal study materials (lines 40-43). **Therefore, it is reasonable to assume that each manuscript was written by a single writer.**

---

> > ### Author Response · Authors · 2025-11-22
> > **Official Comment by Authors to Reviewer aFsT (Questions)**
> >
> > We thank the reviewer for the feedback and comments. Our responses to the questions are given below.
> >
> > 1.      Since adapters are composed additively, have the authors tested whether the order of adapter application (style -> language -> task vs. the reverse) affects convergence or performance?
> > Response: **The adapters are integrated through element-wise addition of their weights.** Since this composition is purely additive, the **order of integration has no effect** on convergence or performance.
> >
> > 2.      What are the total training times and GPU hours for AdapterTrOCR and DiffLine? How do they compare with full fine-tuning and prior PEFT baselines?
> > Response: Training DiffLine requires 41 GPU hours. Training AdapterTrOCR takes 29.6 GPU hours, compared to 27.8 GPU hours for TrOCR. AdapterTrOCR additionally uses four adapters, each requiring an average of 5.5 GPU hours to train; however, these adapters can be reused when training new AdapterTrOCR models (lines 1042-1045 in Appendix A8).
> >
> > 3.      Could the adapter composition strategy generalize to other low-resource scripts (e.g., Old French, Medieval Spanish)? Have preliminary tests been done?
> > Response: **We show that AdapterTrOCR can outperform TrOCR not only for Latin but also for other languages.** We prove this on two additional HTR datasets written in German and French. Further details are provided above in our response to Reviewer 2uJK (in paper: lines 160-161 and Appendix A2).

---

### Official Review · Reviewer_2uJK · 2025-11-03

**Soundness:** 1
**Presentation:** 2
**Contribution:** 1
**Rating:** 2
**Confidence:** 5

**Summary:**

This article proposes a method for adapting the TrOCR model to historical documents in a language with limited resources. The proposed method is based on two independently conducted adaptations: one to historical writing and one to language. An evaluation is conducted by adapting a TrOCR model trained for modern English to documents in historical Latin.

**Strengths:**

The article puts forward the interesting hypothesis that handwriting recognition models can be adapted independently of document style (historical) and language. If such an approach were validated, it would enable models to be adapted to a wider variety of documents and languages, taking advantage of combinations for which little training data is available.

**Weaknesses:**

While the hypothesis is interesting, the experimental protocol and datasets used do not allow it to be validated:

- on the dataset used, in scenario 2, TrOCR already performs quite well: 4.47% CER. The improvement with the proposed adaptation is therefore small, even if significant. In scenario 1, the model remains underperforming even after adaptation (23% CER to 20% CER).
- the generality of the method has not been demonstrated, as the experiments were conducted on a single historical dataset (VOC) and a single language (Latin). Many other historical datasets exist, so other combinations could be tested.
- the proposed method is not compared to a standard alternative, which would consist of adapting the TrOCR model to the historical dataset and then adapting it to the target language.

The use of synthetic data somewhat obscures the message of the article, which should focus on its initial hypothesis: model adaptation.

The method could be tested on larger models, for which LoRA was originally developed, such as Qwen3.

The details of the LoRA parameters are not provided.

**Questions:**

L040 : "written in Latin and date from the 16th to 18th centuries" : It is unusual to find manuscripts from this period written in Latin. You should describe the documents.

L044 : "HTR is usually implemented as a two-step process" :  more and more approaches are based on full-page models (DAN, vLLM).

L62 : "Since the simple fine-tuning is not sufficient" : explain why and show results.

L349 : accuracy is not defined : how is it computed, is it at character level ?

---

> ### Author Response · Authors · 2025-11-22
> **Official Comment by Authors to Reviewer 2uJK (First part)**
>
> We thank the reviewer for the feedback and comments. Our responses to weaknesses 1-4 are given below.
> 1.      on the dataset used, in scenario 2, TrOCR already performs quite well: 4.47% CER. The improvement with the proposed adaptation is therefore small, even if significant. In scenario 1, the model remains underperforming even after adaptation (23% CER to 20% CER).
> Response: We acknowledge the reviewer’s observation regarding the apparent magnitude of the improvements. However, the gains are in fact substantial when they are examined in relative terms. In **Scenario 1 (poorly represented manuscript), our model AdapterTrOCR decreases CER and WER by 15.3% and 8.1%, while improving the accuracy by 7.08%; in Scenario 2 (well-represented manuscript), as a result of applying AdapterTrOCR, CER and WER decrease by 46.6% and 29.7%, with 20.42% higher accuracy**. These results show that our method provides meaningful gains across both scenarios.
>
> 2.      the generality of the method has not been demonstrated, as the experiments were conducted on a single historical dataset (VOC) and a single language (Latin). Many other historical datasets exist, so other combinations could be tested.
> Response: While the primary goal of our project is to transcribe Latin manuscripts held by a European library (lines 40–43), **we also evaluate fine-tuned TrOCR and AdapterTrOCR on French (16th–18th centuries) and German (16th century) datasets to demonstrate the generalization of our approach. AdapterTrOCR consistently outperforms fine-tuned TrOCR, although the improvements are modest because our model includes only the historical-language adaptation. Since both the French and German datasets lack writer information, the writing-style adaptation and handwriting-style-specific data augmentation could not be applied (lines 160-161 and Appendix A2)**.
> |                | French Dataset |        |        | German Dataset |        |        |
> |----------------|----------------|--------|--------|----------------|--------|--------|
> | **Method**     | **ACC**        | **CER**| **WER**| **ACC**        | **CER**| **WER**|
> | TrOCR          | **10.53**          | 18.73  | 47.64  | 8.73           | 25.64  | 49.75  |
> | AdapterTrOCR (historical language adaptation) | 10.49 | **16.98** | **46.89** | **9.07** | **24.12** | **49.09** |
>
> 3.      the proposed method is not compared to a standard alternative, which would consist of adapting the TrOCR model to the historical dataset and then adapting it to the target language
> Response: During our preliminary tests, we tried the suggested setup (TrOCR + Historical Dutch HTR + Latin CLM Adapter). **However, we found that, instead of simply adding the target language adapter, it is more effective to add the language difference between Latin and the proxy language (Dutch in our case). Using the language difference explicitly forces the model to “unlearn” Dutch-like language features introduced by the historical HTR adapter, which is trained on Dutch manuscripts (lines 461-465)**. As shown below, adding the language difference yields better results than just adding the Latin adapter:
> |              | 1st scenario |            |        | 2nd scenario |            |        |
> |--------------|--------------|------------|--------|--------------|------------|--------|
> | **Method**   | **ACC**      | **CER**    | **WER**| **ACC**      | **CER**    | **WER**|
> | TrOCR        | 4.33         | 23.32      | 60.03  | 37.79        | 4.47       | 13.59  |
> | TrOCR + (Historical Dutch HTR + Latin CLM Adapter) | 4.00 | 23.12 | 60.45 | 38.53 | **4.03** | 13.40 |
> | TrOCR + historical Latin adapter (Ours) | **4.00** | **22.73** | **60.34** | **40.80** | 4.15 | **12.88** |
>
> 4.      The use of synthetic data somewhat obscures the message of the article, which should focus on its initial hypothesis: model adaptation.
> Response: While we use synthetic data to augment the training data of AdapterTrOCR, **the main benefit of data augmentation is to enhance the training data of the handwriting-style adapter, which is a key component of our model adaptation (as shown by the results in Table 3)**.

---

> > ### Author Response · Authors · 2025-11-22
> > **Official Comment by Authors to Reviewer 2uJK (Second part)**
> >
> > We thank the reviewer for the feedback and comments. Our responses to weaknesses 5-6 and the questions 1-4 are given below.
> >
> > **Weaknesses (5-6):**
> >
> > 5.      The method could be tested on larger models, for which LoRA was originally developed, such as Qwen3.
> > Response: While the reviewer is right and larger vision–language models than TrOCR do exist, **we selected TrOCR as a backbone for our model because it is already pre-trained for HTR**. To justify this choice, **we compare TrOCR with Qwen2**, for which an OCR-pretrained checkpoint is available (we couldn’t find an OCR-based checkpoint for Qwen3), **and with Gemma** (a model inspired by Gemini which is suggested by Reviewer UKgu). **All three models are fine-tuned under the second scenario, which targets a manuscript that is well represented in the training data. The results show that TrOCR’s HTR-specific pretraining provides in general a better advantage for the Latin HTR task than larger vision–language models with limited or nonexistent prior HTR exposure** (lines 151-152 and Appendix A1). The results are provided for the second scenario (well-represented manuscript).
> > | Method | ACC   | CER   | WER   |
> > |--------|-------|-------|-------|
> > | Qwen2  | 37.24 | **4.23**  | 14.11 |
> > | Gemma  | 35.87 | 6.63  | 16.43 |
> > | TrOCR  | **37.79** | 4.47  | **13.59** |
> >
> > 6.      The details of the LoRA parameters are not provided.
> > Response: the training setup is provided in Appendix A.8. The details of the LoRA parameters are provided between lines 1024-1026.
> >
> > **Questions:**
> >
> > 1.      L040 : "written in Latin and date from the 16th to 18th centuries" : It is unusual to find manuscripts from this period written in Latin. You should describe the documents.
> > Response: While Latin was no longer a spoken language between the 16th and 18th centuries, it remained widely used in academia and education. Since our collection consists of student notes from various disciplines, covering topics from logic to physics, the language used is Latin (lines 40-43). Due to the ICLR’s anonymity policy, we cannot provide more details.
> >
> > 2.      L044 : "HTR is usually implemented as a two-step process" : more and more approaches are based on full-page models (DAN, vLLM).
> > Response: **We have reformulated this statement (lines 44–46). For our methodology, we stick with the two-step approach, as the YOLO model fine-tuned for line detection is already highly effective, allowing us to focus exclusively on line-level HTR**. Additionally, we have already compared our method (integrated with YOLO) with the end-to-end HTR model ViTLP, obtaining better results (Table 2). Our evaluation is limited to ViTLP because, unlike DAN, which uses a CNN backbone for visual feature extraction, ViTLP provides a fully transformer-based encoder-decoder model. On the other hand, vLLM does not offer an HTR-specific checkpoint.
> >
> > 3.      L62 : "Since the simple fine-tuning is not sufficient" : explain why and show results.
> > Response: Whenever we compare TrOCR (or any other model) with AdapterTrOCR, all models are fine-tuned on the same dataset. Therefore, simply fine-tuning TrOCR on our Latin HTR data is not sufficient and the additional adaptation provided by our method is necessary to obtain a model that is truly effective for Latin HTR. This explanation is outlined in lines 346-347 and lines 353-354.
> >
> > 4.      L349 : accuracy is not defined : how is it computed, is it at character level ?
> > Response: Accuracy evaluates how often the model’s transcription exactly matches the ground truth; this definition is provided in lines 341–342 and 384-386.

---

### Meta-Review · Area_Chair_P7r2 · 2026-01-07

**Summary:**

This paper proposes a method for adapting TrOCR to historical documents in a language with limited resources. The authors propose two adapter modules: one for adapting to the historical Latin language and the other for adapting to specific handwriting styles. Overall, the paper is well motivated and easy to follow.

**Reviewer Concerns:**

The concerns are around lack of novelty, lack of comparison methods, limited improvement, generlizability to other datasets/languages, etc. The authors have provided some additional experimental results in the rebuttal and tried to argue with the reviewers. However, some major concerns as follows have not been fully addressed:

[Reviewer 2uJK/Vj1a/UKgu] "The generality of the method has not been demonstrated, as the experiments were conducted on a single historical dataset (VOC) and a single language (Latin)." The authors cited the results from Appendix A4 on smaller French and German datasets in the rebuttal, which may not well support the generality of the proposed method.

[Reviewers 2uJK/Vj1a] Limited performance improvement.

[Reviewers 2uJK/Vj1a] Incremental novelty. Many core components rely on adapting existing architectures (e.g., TrOCR, DiffusionPen) via LoRA fine-tuning or different existing loss functions.

**Reviewer Scores:**

Reviewers 2uJK and Vj1a are unlikely to change their scores (both rated 2) as their major concerns are exactly the abovementioned unsolved major concerns (especially Reviewers 2uJK with Confidence 5).

Reviewer aFsT might slightly raise the score because his/her concerns are relatively minor or requests on details.

It is hard to predict Reviewer UKgu's score becuase his/her major concerns also largely overlap with the abovementioned unsolved major concern (however with Confidence 3).

---

### Decision · Program_Chairs · 2026-01-26

Reject